# Effectiveness of Nootropics in Combination with Cholinesterase Inhibitors on Cognitive Function in Mild-to-Moderate Dementia: A Study Using Real-World Data

**DOI:** 10.3390/jcm11164661

**Published:** 2022-08-09

**Authors:** Minjae Kang, Dan Bee Lee, Sungchan Kwon, Eun Lee, Woo Jung Kim

**Affiliations:** 1Department of Psychiatry, Severance Hospital, Yonsei University College of Medicine, Seoul 03722, Korea; 2Medical Information Team, Yongin Severance Hospital, Yonsei University College of Medicine, Yongin 16995, Korea; 3Division of Gastroenterology, Department of Internal Medicine, Severance Hospital, Yonsei University College of Medicine, Seoul 03722, Korea; 4Institute of Behavioral Sciences in Medicine, Yonsei University College of Medicine, Seoul 03722, Korea; 5Department of Psychiatry, Yongin Severance Hospital, Yonsei University College of Medicine, Yongin 16995, Korea

**Keywords:** cholinesterase inhibitors, nootropic agents, dementia, Alzheimer disease, cognitive dysfunction, mental status and dementia test

## Abstract

The clinical benefits of nootropics in the treatment of cognitive decline has been either limited or controversial. This study aimed to observe the effectiveness of cholinesterase inhibitor (ChEI) and nootropics combination in the treatment of cognitive impairment in dementia. Data were based on electronic medical records in a university health system. Patients with mild-to-moderate dementia and no history of prior cognitive enhancer use were included (*n* = 583). The subjects were categorized into the ChEI only group and the ChEI and nootropics combination group. The primary outcome measure was the change in cognitive function, as assessed by the mini-mental state examination (MMSE) from baseline to 300–400 days after the first ChEI prescription. Subsequent analyses were conducted in consideration of the dementia type, medical adherence, and type of nootropics. The changes in MMSE scores from baseline to endpoint were not significantly different between the two groups. In Alzheimer’s dementia, the combination group showed significantly less deterioration in MMSE language subscale scores compared to the ChEI only group (F = 6.86, *p* = 0.009), and the difference was consistent in the highly adherent subjects (F = 10.16, *p* = 0.002). The choline alfoscerate and the ginkgo biloba extract subgroups in Alzheimer’s dementia showed more significant improvements in the MMSE language subscale scores compared to the other nootropics subgroup (F = 7.04, *p* = 0.001). The present study showed that the effectiveness of ChEI and nootropics combination on cognition may appear differently according to the dementia type. This emphasizes the need for well-controlled studies to generalize the effectiveness of nootropics across various clinical settings.

## 1. Introduction

Dementia is one of the most rapidly growing syndromes in aging societies, and it is also the leading cause of disability and caregiver dependency among older adults. Currently, more than 55 million people worldwide are affected by dementia, and nearly 10 million new patients are diagnosed with dementia every year [1]. Dementia leads to deterioration in multiple domains of cognitive function, including orientation, memory, language, attention, and visuospatial construction. To date, no curative drugs for dementia have been widely implemented in clinical practice, and most clinicians still consider cholinesterase inhibitors (ChEIs) as the first line of pharmacological treatment for cognitive deficits in dementia [2,3].

Nootropics, which were originally described as compounds with similar pharmacology to piracetam, are now generally accepted as any and all compounds that are expected to improve learning and memory in experimental and clinical paradigms [4]. In Korea, apart from the general over-the-counter supplements, there are certain nootropics that need prescriptions for use; among those most frequently prescribed are choline alfoscerate, ginkgo biloba extract, acetyl-L-carnitine, nicergoline, and oxiracetam. Many clinicians often prefer concomitant use of ChEIs with nootropics, based on fragmentary evidence that augmentation could have certain clinical benefits [5,6]. In fact, the clinical benefits of nootropics in the treatment of cognitive decline are either limited or controversial, with some studies showing improvements in cognitive function with choline alfoscerate [7,8,9], ginkgo biloba extract [10,11], acetyl-L-carnitine [6,12], and nicergoline [13,14], while other studies demonstrated no clinical benefits of such nootropics [15,16,17,18]. Although nootropics alone may not significantly improve cognitive function, nootropics in combination with ChEIs may enhance cholinergic neurotransmission, neurogenesis, and micro-perfusion in the frontal lobe, hippocampus, and striatum, ultimately delaying cognitive decline in dementia, especially in those with mild-to-moderate disease severity [19,20,21].

This study aimed to elucidate whether the concomitant use of nootropics with ChEIs is effective in the treatment of cognitive impairment in patients with dementia. We hypothesized that dementia patients treated with both ChEIs and nootropics would show less deterioration in cognitive function as assessed by mini-mental state examination (MMSE) over a 1-year follow-up period compared to dementia patients treated with ChEIs only. We further hypothesized that this trend would be more evident in the neurodegenerative forms of dementia, particularly Alzheimer’s dementia.

## 2. Materials and Methods

### 2.1. Study Population

The present study aimed to observe the effectiveness of the concomitant administration of nootropics with ChEIs for the treatment of cognitive deficits in patients with dementia. Data were collected and extracted via the clinical data warehouse of a university health system consisting of three general hospitals with 4050 beds in total. Patients with any diagnosis of dementia, defined according to the International Statistical Classification of Disease and Related Health Problems 10th Revision (ICD-10) diagnostic codes F00, F01, F02, F03, F10.7, G23.1, G30, G31.0, G31.1, and G31.8, between the ages of 60 and 89 years were selected. We extracted all diagnostic codes of the patients regardless of whether they were the main or sub-diagnoses. After screening the entire prescription histories registered on the clinical data warehouse, which was over 7 years of record, only the patients who had their first ChEI prescription between 2012 and 2015 were included in the study. Subjects were further narrowed down to only those with mild-to-moderate dementia, as defined by their baseline MMSE total scores from 10 to 26 (*n* = 583) (Figure 1). Subjects were categorized into the ChEI only group (*n* = 410) and the ChEI and nootropics combination group (*n* = 173). Subjects in the ChEI and nootropics combination group were prescribed at least one of the following nootropics during the follow-up period: choline alfoscerate, ginkgo biloba extract, acetyl-L-carnitine, nicergoline, and oxiracetam. MMSE total scores and its six-subscale scores (orientation, immediate recall, attention and calculation, delayed recall, language, and visuospatial construction) were collected at the baseline (within 30 days of the first ChEI prescription of ChEI) and at the endpoint (300–400 days after the first ChEI prescription), respectively. Baseline data on age, sex, weight, height, body mass index (BMI), blood pressure (BP), education, alcohol use, tobacco use, concomitant psychiatric medications, and comorbid diagnoses were collected during the study period. Subjects were categorized by the type of dementia according to their main diagnosis in the highest order: Alzheimer’s dementia (*n* = 447), vascular dementia (*n* = 80), and other dementia (*n* = 56). For subgroup analysis, the ChEI and nootropics combination group of the Alzheimer’s dementia patients were further classified as follows: choline alfoscerate group (*n* = 74), ginkgo biloba extract group (*n* = 39), and other nootropics group (*n* = 21). Other nootropics included acetyl-L-carnitine, nicergoline, and oxiracetam.

This research protocol was exempt from an ethics review and informed consent was waived by the Institutional Review Board of Yongin Severance Hospital (9-2020-0108). This study adhered to the doctrine of the Declaration of Helsinki for Biomedical Research.

### 2.2. Measures

The main outcome measure was the change in cognitive function from baseline to endpoint, as assessed by the MMSE total score and its six subscale scores. Since the primary analysis did not differentiate the dementia type, subsequent analysis was conducted by comparing the changes in cognitive function between the ChEI only and ChEI and nootropics combination groups for each type of dementia separately. Further analysis was conducted in consideration of adherence to ChEI and nootropics. We defined the proportion of days covered (PDC = (Number of days in period “covered” by prescription) × 100/(Number of days in period)) ≥ 0.7 to be highly adherent, based on previous studies on adherence [22]. In the subgroup analysis, MMSE score changes were analyzed among the subgroups (i.e., choline alfoscerate vs. ginkgo biloba extract vs. other nootropics) to observe the effectiveness of each nootropic type.

### 2.3. Statistical Analysis

Baseline demographic and clinical characteristics between the two groups were compared by independent *t*-test for continuous variables and χ2 test or Fisher’s exact test for categorical variables. Two-sample *t*-tests were utilized to compare the two treatment groups at baseline and endpoint, as well as the change from baseline to endpoint. In order to adjust for critical covariates such as time and other relevant factors, including sex, age, hypertension, diabetes mellitus, stroke, Parkinsonism, visual disturbance/hearing loss, hemiplegia/paraplegia, mood/anxiety disorder, we further used linear mixed-effect model analyses with time, treatment group, and their interaction to test the difference of the change in MMSE total score and its six-subscale scores from baseline to endpoint between the two treatment groups. The analysis of variance (ANOVA) with Bonferroni multiple comparisons test was conducted to compare the changes in MMSE total score and its six-subscale scores among the three ChEI and nootropics combination subgroup of Alzheimer’s dementia (i.e., choline alfoscerate vs. ginkgo biloba extract vs. other nootropics).

All statistical analyses were conducted using IBM SPSS ver. 26 for Windows (IBM Corp: Armonk, NY, USA), and statistical significance was determined at *p* < 0.05 (two-tailed).

## 3. Results

### 3.1. Demographic and Clinical Characteristics

Table 1 shows the comparison of demographic and clinical characteristics of the 583 subjects included in this study. The mean ages were similar in both groups (75.5 ± 6.5 years vs. 75.5 ± 6.6 years in the ChEI only group and the ChEI and nootropics combination group, respectively). Females outnumbered males in both groups (266 (64.9%) females vs. 144 (35.1%) males in the ChEI only group; 117 (67.6%) females vs. 56 (32.4%) males in the ChEI and nootropics combination group). Regarding dementia diagnoses, Alzheimer’s dementia was the most frequent in both groups (313 (76.3%) in the ChEI only group, 134 (77.5%) in the ChEI and nootropics combination group), and vascular dementia was the second most frequent (49 (12.0%) in the ChEI only group, 31 (17.9%) in the ChEI and nootropics combination group). Alcohol use was more frequent in the ChEI only group (85 (20.7%)) than in the ChEI and nootropics combination group (23 (13.3%)). There was no difference in the distribution of the type of ChEI prescribed between the two groups (*p* = 0.446 for donepezil, *p* = 0.334 for rivastigmine, *p* = 0.092 for galantamine) (Appendix A).

### 3.2. Change in Cognitive Function: ChEI Only vs. ChEI and Nootropics Combination Group

Results for the primary outcome measure are shown in Table 2. Two-sample *t*-tests at baseline and endpoint showed no difference between the two groups in terms of the MMSE total score and any of its six subscale scores. A linear mixed-effects model analysis with the MMSE total score as the dependent variable also revealed no significant treatment-by-time interaction (F = 1.48, df = 1581, *p* = 0.224). None of the six-subscale scores showed any significant treatment-by-time interaction, albeit the change in language subscale score showed a trend of less deterioration in the nootropics combination group than in the ChEI only group (−0.27 for the ChEI only group and −0.03 for the ChEI and nootropics combination group; F = 3.43, df = 1581, *p* = 0.065).

### 3.3. Change in Cognitive Function for Each Type of Dementia: Alzheimer’s Dementia, Vascular Dementia, and Other Dementia

Comparison of the change in cognitive function for each of the three types of dementia is shown in Table 3. Linear mixed-effects model analyses using the MMSE total score as the dependent variable revealed no significant treatment-by-time interaction in all three types of dementia (F = 2.10, *p* = 0.148 for Alzheimer’s dementia, F = 0.38, *p* = 0.540 for vascular dementia, F = 0.04, *p* = 0.906 for other dementia). Within the subjects diagnosed with Alzheimer’s dementia, the ChEI and nootropics combination group showed a significantly less deterioration in the MMSE language subscale score compared to the ChEI only group, revealing a significant treatment-by-time interaction (−0.34 for the ChEI only group and 0.04 for the ChEI and nootropics combination group; F = 6.86, *p* = 0.009), while in the vascular dementia subjects, the ChEI and nootropics combination group showed a significantly less deterioration in the MMSE attention and calculation subscale score compared to the ChEI only group, with a significant treatment-by-time interaction (−0.59 for the ChEI only group and 0.06 for the ChEI and nootropics combination group; F = 4.44, *p* = 0.038). None of the six-subscale scores showed any significant treatment-by-time interaction in the other dementia patients.

For highly-adherent Alzheimer’s dementia patients, the MMSE language subscale score consistently revealed a significant treatment-by-time interaction, showing favorable results for the ChEI and nootropics combination group (−0.38 for the ChEI only group and 0.12 for the ChEI and nootropics combination group; F = 10.16, *p* = 0.002) (Table 4). Apart from the language subscale score, a linear mixed-effects model with the MMSE visuospatial construction subscale score as the dependent variable also showed a significant treatment-by-time interaction (−0.06 for the ChEI only group and 0.06 for the ChEI and nootropics combination group; F = 4.00, *p* = 0.046) (Table 4).

### 3.4. Subgroup Analysis within the ChEI and Nootropics Combination Group in Alzheimer’s Dementia: Choline Alfoscerate vs. Ginkgo Biloba Extract vs. Other Nootropics

When the three types of nootropics within the ChEI and nootropics combination group of the Alzheimer’s dementia subjects were compared, choline alfoscerate and ginkgo biloba extract combination with ChEI showed significantly less declines in the MMSE total scores (*p* = 0.006) compared to other nootropics combination (−0.28 for the choline alfoscerate group, −0.51 for the ginkgo biloba extract group, −2.81 for the other nootropics group) (Appendix A). Highly significant between-group differences, in favor of the choline alfoscerate combination and the ginkgo biloba extract combination, were shown in the MMSE language subscale score (*p* = 0.001) as well (0.22 for the choline alfoscerate group, 0.26 for the ginkgo biloba group, −0.95 for the other nootropics group) (Appendix A).

## 4. Discussion

Our study findings suggest that the effectiveness of nootropics combination with ChEI may be different according to the type of dementia. The difference in change in the MMSE total scores from baseline to endpoint between the ChEI only group and the ChEI and nootropics combination group was not evident, but the results implied a positive impact of ChEI augmentation with nootropics on some domains of cognitive function in certain types of dementia, particularly Alzheimer’s dementia. The impact of nootropics augmentation on ChEI in Alzheimer’s dementia was consistently depicted on the language domain after consideration of medical adherence and nootropics type.

Previous studies on nootropics revealed nootropics to have minimal, if any, effects on the prevention of cognitive decline. Most studies were either examined over a relatively short follow-up time or conducted with a limited number of subjects, and many of them focused on nootropic monotherapy rather than concomitant use with ChEI [6,9,10,17,23,24,25]. This study aimed to compare the nootropics and ChEI combination with ChEI monotherapy in an attempt to validate the effectiveness of nootropics in association with ChEI. We examined the change in not only MMSE total scores but also in its six subscale scores over a long period of exposure (300–400 days after the first prescription), which was considered long enough to elucidate the effect of the ChEI and nootropics combination. We categorized the subjects by dementia types in order to clarify the impact of nootropics combination in each type of dementia separately, and mainly focused on Alzheimer’s dementia, which is the most common neurodegenerative form of dementia. We did not limit nootropics to only one type of drug, but included multiple types of nootropics that are most frequently prescribed by clinicians in Korea, to speculate the effectiveness of different types of nootropics, particularly focusing on the two drugs that were most commonly used: choline alfoscerate and ginkgo biloba extract.

Choline alfoscerate is a derivative of phosphatidylcholine that enhances cholinergic transmission by upregulating acetylcholine (ACh) synthesis or release in the hippocampus [26,27]; thus, it ultimately facilitates learning and memory and decreases age-dependent structural changes, as seen in the frontal cortices and hippocampi of rats [7,28]. Previous studies reported that the combination of choline alfoscerate with ChEI significantly increased ACh concentrations and prevented volume loss in the frontal and temporal lobe, hippocampus, and striatum in both rats [20] and in Alzheimer’s dementia patients [21]. One study also showed that choline alfoscerate increased hippocampal neurogenesis and provided protection against seizure-induced neuronal death and cognitive impairment [29]. A study on the supplementary effect of choline alfoscerate on speech detection and recognition among hearing aid users revealed that in the aging brain, nicotinic acetylcholine receptor activation in the medial geniculate body decreases, which contributes to deterioration in speech recognition and comprehension and, in such cases, choline precursor supplements could improve language functioning [30]. Ginkgo biloba extract, on the other hand, is hypothesized to enhance amyloid β-induced hippocampal neuron dysfunction and death, amyloid β aggregation, and neurogenesis [31,32,33], and induce a significant decrease in the density of β-adrenoreceptors in the frontal cortex and hippocampus [34]. Studies have shown that the hippocampus is closely associated with language production and verbal communication, either by contributing semantic memory to spoken language, processing the mismatch between the expected sensory consequences of speaking and perceived speech feedback, or via coupling between the hippocampal/supplementary motor area and the auditory cortex, anterior cingulate cortex, and cerebellum [30,35]. These findings all suggest that nootropics, such as choline alfoscerate and ginkgo biloba extract, may enhance the cholinergic pathway in brain areas including the hippocampus, frontal lobes, and auditory cortex, which are critical in language processing and utilization, and enhance language function in association with ChEI administration. Language function is not only highly sensitive in detecting cognitive impairment in older age groups [36], but is also a biological marker that differentiates Alzheimer’s dementia from normal aging [37]. To our knowledge, this study is the first to elucidate the effect of nootropics co-administered with ChEIs on the attenuation of language domain deterioration in Alzheimer’s dementia using clinical data. The possible effect of nootropics on delaying the deterioration of language function may be of great value, considering the association between cognitive decline and language function in the aging population and dementia patients.

The strength of this study lies in the methodological aspects of real-world data utilization. Many studies in Korea are actively conducted using the national claims database in an attempt to perform large-scale research [38]. However, imperative baseline characteristics of each patient, including medical histories, medication, and laboratory findings, are missing when using the national claims databases. On the other hand, the clinical data warehouse from our university healthcare system is a composition of each patient’s clinical and demographic data. In this study, we were able to extract and collect critical information regarding each patient’s clinical and demographic backgrounds such as, but not limited to, the education level and BMI, specific MMSE subscale scores, and detailed dementia diagnosis. As a result, we were able to make corrections for many variables that could have affected the MMSE results, including other relevant diagnoses and medical history, such as hypertension and diabetes mellitus, Parkinsonism, and visual/hearing loss. Furthermore, this study is meaningful in that until now, there have only been a few, if any, studies on the effect of nootropics on ChEI efficacy outside of the only actively researched country, Italy [19,20,39].

The study had some limitations. Due to its retrospective nature, we were not able to control the treatment settings in terms of medication dosage, treatment duration, the overlapping periods of concomitant nootropic and ChEI use, and adherence monitoring. In order to compensate for these factors, we calculated and considered PDC, the number of days covered by the prescriptions in the total prescription periods, to reflect and consider drug adherence. Moreover, we only included subjects with MMSE scores within 300–400 days after the baseline, which might have excluded certain patients who could not take the annual MMSE test either due to a deleterious progression of dementia or because they no longer sought help or changed hospitals. In the near future, it seems that these limitations can be solved by conducting research based on a large database that combines real-world data from numerous hospitals (i.e., common data model). Finally, analyzing the six subscale domains of the MMSE may raise the issue of multiple comparison. However, through subsequent analyses, we induced consistent results at least in the Alzheimer’s dementia group, of which nootropics combination could have a subtle positive impact on the language domain. Surely, further studies that more deliberately focus on the separate cognitive domains should be performed with other more qualified neuropsychological tests. Still, this study would be of best effort to observe the effectiveness of nootropics utilizing the accumulated real-world data.

## 5. Conclusions

In conclusion, our results demonstrated partial effectiveness of nootropics with ChEIs on some cognitive domains in Alzheimer’s dementia. The reliability of the current study was further strengthened by its 1-year follow-up period, consideration of treatment adherence, and adjustments for confounding comorbid diagnoses. This study emphasizes the need for future well-controlled studies to generalize the effect of nootropics on cognitive function across various clinical settings or cause of dementia.

## Figures and Tables

**Figure 1 jcm-11-04661-f001:**
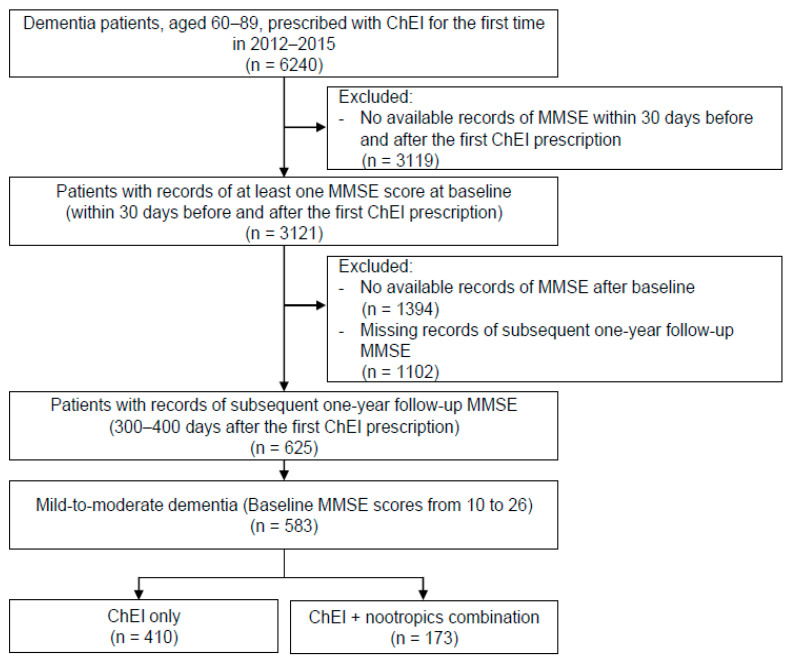
Flow diagram of study sample selection. ChEI: cholinesterase inhibitor; MMSE: mini-mental state examination.

**Table 1 jcm-11-04661-t001:** Demographic and clinical characteristics of the cholinesterase inhibitor only group and the nootropics combination group.

Variable	ChEI Only (*n* = 410)	ChEI + Nootropics (*n* = 173)	*p* Value
Age (years)	75.5 ± 6.5	75.5 ± 6.6	0.880
Weight (kg) ^1^	56.4 ± 9.8	56.7 ± 12.2	0.843
Height (cm) ^2^	157.3 ± 10.1	156.1 ± 11.5	0.274
BMI (kg/m^2^) ^3^	22.8 ± 3.8	23.3 ± 3.8	0.265
Blood pressure (mmHg) ^4^			
Systolic	130.3 ± 20.8	129.1 ± 22.1	0.576
Diastolic	75.6 ± 12.9	75.6 ± 12.9	0.962
Sex			
Male	144 (35.1)	56 (32.4)	0.523
Female	266 (64.9)	117 (67.6)	
Education (years) ^5^			
No education	55 (13.4)	20 (11.6)	0.651
Elementary school	94 (22.9)	43 (24.9)	
Middle/high school	90 (21.9)	31 (7.9)	
Bachelor’s/master’s degree	64 (15.6)	30 (17.3)	
Alcohol use ^6^			
Yes	85 (20.7)	23 (13.3)	0.011
No	203 (49.5)	107 (61.8)	
Tobacco use ^7^			
Yes	23 (5.6)	7 (4.0)	0.245
No	224 (54.6)	114 (65.9)	
Psychiatric medication			
Antipsychotics	73 (17.8)	34 (19.7)	0.598
Benzodiazepines, hypnotics	119 (29.0)	45 (26.0)	0.460
Anticholinergics	83 (20.2)	43 (24.9)	0.217
Dementia diagnoses			
Alzheimer’s dementia	313 (76.3)	134 (77.5)	0.771
Vascular dementia	49 (12.0)	31 (17.9)	0.056
Unspecified dementia	17 (4.1)	6 (3.5)	0.701
Frontotemporal dementia	8 (2.0)	0 (0)	0.113
Progressive supranuclear palsy	16 (3.9)	0 (0)	0.004
Alcohol induced persisting dementia	1 (0.2)	0 (0)	1.000
Other specified degenerative disease of nervous system	6 (1.5)	2 (1.2)	1.000
Comorbid diagnoses			
Diabetes mellitus	81 (19.8)	50 (28.9)	0.016
Hypertension	138 (33.7)	70 (40.5)	0.117
Hyperlipidemia	55 (13.4)	29 (16.8)	0.293
Stroke ^8^	36 (8.8)	36 (20.8)	<0.001
Parkinsonism	69 (16.8)	12 (6.9)	0.002
Visual disturbance/hearing loss	26 (6.3)	20 (11.6)	0.033
Hemiplegia/paraplegia	3 (0.7)	6 (3.5)	0.023
Asthma/COPD	18 (4.4)	11 (6.4)	0.318
Renal insufficiency	25 (6.1)	14 (8.1)	0.378
Mood/anxiety disorder	141 (34.4)	75 (43.4)	0.041
Other psychiatric disorder ^9^	26 (6.3)	12 (6.9)	0.790
Any malignancy	26 (6.3)	10 (5.8)	0.797

Values are presented as mean ± standard deviation or number (%). Independent *t*-test was used for continuous variables, and χ2 test or Fisher’s exact test was used for categorical variables. ChEI: cholinesterase inhibitor; BMI: body mass index; COPD: chronic obstructive pulmonary disease. ^1^ Missing: ChEI only = 132, ChEI + nootropics = 51. ^2^ Missing: ChEI only = 136, ChEI + nootropics = 53. ^3^ Missing: ChEI only = 81, ChEI + nootropics = 33. ^4^ Missing: ChEI only = 102, ChEI + nootropics = 40. ^5^ Missing: ChEI only = 107, ChEI + nootropics = 49. ^6^ Missing: ChEI only = 122, ChEI + nootropics = 43. ^7^ Missing: ChEI only = 163, ChEI + nootropics = 52. ^8^ Cerebral hemorrhage and ischemic stroke. ^9^ Psychotic disorder and substance use disorder.

**Table 2 jcm-11-04661-t002:** Results of significance testing and mixed model analyses for mini mental state examination (MMSE): Cholinesterase Inhibitor Only vs. Cholinesterase Inhibitor and Nootropics Combination ^1^.

	Baseline			Endpoint			Difference in Scores			Group-Time Interaction ^2^	
MMSE	ChEI Only	ChEI + Nootropics	*p* Value ^3^	ChEI Only	ChEI + Nootropics	*p* Value ^3^	ChEI Only	ChEI + Nootropics	*p* Value ^3^	F	*p* Value ^4^
Total	20.00 ± 4.30	20.27 ± 4.26	0.487	18.87 ± 5.43	19.56 ± 5.13	0.152	−1.13 ± 3.98	−0.71 ± 3.47	0.199	1.48	0.224
Orientation	6.81 ± 2.33	6.94 ± 2.23	0.533	6.28 ± 2.70	6.54 ± 2.57	0.276	−0.53 ± 2.21	−0.40 ± 2.15	0.504	0.45	0.504
Immediate recall	2.83 ± 0.47	2.84 ± 0.47	0.925	2.74 ± 0.63	2.77 ± 0.59	0.526	−0.10 ± 0.66	−0.06 ± 0.63	0.594	0.28	0.594
Attention & calculation	1.99 ± 1.58	2.01 ± 1.57	0.881	1.81 ± 1.58	1.95 ± 1.62	0.327	−0.18 ± 1.53	−0.06 ± 1.29	0.364	0.83	0.364
Delayed recall	0.92 ± 1.00	1.10 ± 1.11	0.064	0.90 ± 1.06	0.93 ± 1.10	0.715	−0.03 ± 1.02	−0.17 ± 1.16	0.150	2.29	0.131
Language	6.97 ± 1.22	6.87 ± 1.41	0.387	6.70 ± 1.46	6.84 ± 1.55	0.295	−0.27 ± 1.45	−0.03 ± 1.43	0.065	3.43	0.065
Visuospatial construction	0.47 ± 0.50	0.50 ± 0.50	0.561	0.44 ± 0.50	0.51 ± 0.50	0.111	−0.03 ± 0.56	0.01 ± 0.58	0.375	0.79	0.375

Values are presented as mean ± standard deviation. MMSE: mini-mental state examination; ChEI: cholinesterase inhibitor. ^1^ Cholinesterase inhibitor only [*n* = 410] and cholinesterase inhibitor and nootropics combination [*n* = 173]. ^2^ The results for the linear mixed-effects model were adjusted for covariates of sex, age, hypertension, diabetes mellitus, stroke, Parkinsonism, visual disturbance/hearing loss, hemiplegia/paraplegia, and mood/anxiety disorder. ^3^
*p* value for *t*-test. ^4^
*p* value for linear mixed-effects model.

**Table 3 jcm-11-04661-t003:** Results of mixed model analyses ^1^ for mini-mental state examination (MMSE) among the three dementia subgroups.

	Alzheimer’s Dementia (*n* = 447)				Vascular Dementia (*n* = 80)				Others (*n* = 56)			
	Difference in Scores		Group-Time Interaction		Difference in Scores		Group-Time Interaction		Difference in Scores		Group-Time Interaction	
MMSE	ChEI Only (*n* = 313)	ChEI + Nootropics (*n* = 134)	F	*p* Value	ChEI Only (*n* = 49)	ChEI + Nootropics (*n* = 31)	F	*p* Value	ChEI Only (*n* = 48)	ChEI + Nootropics (*n* = 8)	F	*p* Value
Total	−1.30 ± 3.87	−0.75 ± 3.27	2.10	0.148	−0.88 ± 3.46	−0.35 ± 4.06	0.38	0.540	−0.27 ± 5.04	−1.38 ± 4.66	0.04	0.906
Orientation	−0.62 ± 2.16	−0.45 ± 1.91	0.66	0.417	−0.33 ± 2.04	−0.16 ± 3.03	0.09	0.771	−0.15 ± 2.67	−0.50 ± 1.93	0.14	0.707
Immediate recall	−0.08 ± 0.65	−0.09 ± 0.62	0.01	0.922	−0.12 ± 0.81	0.10 ± 0.70	1.55	0.217	−0.15 ± 0.62	−0.25 ± 0.46	0.01	0.937
Attention & calculation	−0.18 ± 1.50	−0.11 ± 1.28	0.21	0.651	−0.59 ± 1.38	0.06 ± 1.32	4.44	0.038	0.25 ± 1.77	0.38 ± 1.41	1.13	0.293
Delayed recall	−0.02 ± 1.05	−0.19 ± 1.15	2.46	0.117	0.00 ± 0.94	−0.13 ± 1.15	0.30	0.584	−0.10 ± 0.99	0.00 ± 1.51	1.35	0.252
Language	−0.34 ± 1.45	0.04 ± 1.39	6.86	0.009	0.14 ± 1.04	−0.10 ± 1.11	0.96	0.331	−0.23 ± 1.72	−1.00 ± 2.67	1.90	0.175
Visuospatial construction	−0.05 ± 0.57	0.05 ± 0.58	3.28	0.071	−0.04 ± 0.54	−0.16 ± 0.53	0.97	0.327	0.56 ± 0.08	0.76 ± 0.27	0.18	0.677

Values are presented as mean ± standard deviation. MMSE, mini-mental state examination; ChEI, cholinesterase inhibitor. ^1^ The results for the linear mixed-effects model were adjusted for covariates of sex, age, hypertension, diabetes mellitus, stroke, Parkinsonism, visual disturbance/hearing loss, hemiplegia/paraplegia, and mood/anxiety disorder.

**Table 4 jcm-11-04661-t004:** Results of mixed model analyses ^1^ for mini-mental state examination (MMSE) for highly-adherent ^2^ Alzheimer’s Dementia Patients.

	Alzheimer’s Dementia (*n* = 412)				Vascular Dementia (*n* = 72)				Others (*n* = 53)			
	Difference in Scores		Group-Time Interaction		Difference in Scores		Group-Time Interaction		Difference in Scores		Group-Time Interaction	
MMSE	ChEI Only (*n* = 300)	ChEI + Nootropics (*n* = 112)	F	*p* Value	ChEI Only (*n* = 45)	ChEI + Nootropics (*n* = 27)	F	*p* Value	ChEI Only (*n* = 45)	ChEI + Nootropics (*n* = 8)	F	*p* Value
Total	−1.36 ± 3.87	−0.77 ± 2.85	2.18	0.141	−0.78 ± 3.51	−0.59 ± 4.17	0.04	0.841	−0.36 ± 5.19	−1.38 ± 4.66	0.27	0.606
Orientation	−0.64 ± 2.16	−0.54 ± 1.82	0.22	0.640	−0.38 ± 1.84	−0.22 ± 3.23	0.07	0.795	−0.18 ± 2.76	−0.50 ± 1.93	0.10	0.753
Immediate recall	−0.08 ± 0.66	−0.07 ± 0.57	0.03	0.865	−0.13 ± 0.84	0.11 ± 0.75	1.54	0.219	−0.16 ± 0.64	−0.25 ± 0.46	0.16	0.692
Attention & calculation	−0.18 ± 1.49	−0.12 ± 1.32	0.16	0.689	−0.58 ± 1.42	0.04 ± 1.22	0.04	0.841	0.13 ± 1.75	0.38 ± 1.41	0.14	0.714
Delayed recall	−0.01 ± 1.03	−0.22 ± 1.11	3.35	0.068	0.07 ± 0.92	−0.19 ± 1.15	1.06	0.307	−0.07 ± 0.99	0.00 ± 1.51	0.03	0.872
Language	−0.38 ± 1.46	0.12 ± 1.26	10.16	0.002	0.20 ± 1.04	−0.14 ± 1.11	2.21	0.142	−0.22 ± 1.74	−1.00 ± 2.67	1.14	0.291
Visuospatial construction	−0.06 ± 0.56	0.06 ± 0.59	4.00	0.046	−0.02 ± 0.54	−0.15 ± 0.53	0.92	0.341	0.13 ± 0.55	0.00 ± 0.76	0.36	0.552

Values are presented as mean ± standard deviation. MMSE, mini-mental state examination; ChEI, cholinesterase inhibitor. ^1^ The results for the linear mixed-effects model were adjusted for covariates of sex, age, hypertension, diabetes mellitus, stroke, Parkinsonism, visual disturbance/hearing loss, hemiplegia/paraplegia, and mood/anxiety disorder. ^2^ Highly-adherent subjects are defined as proportion of days covered ≥ 0.7; Proportion of days covered = (Number of days in the period “covered” by prescription) × 100/(Number of days in period).

## Data Availability

Data are not available according to the policy of Yonsei University College of Medicine.

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
