# Peer review of "Effectiveness of Nootropics in Combination with Cholinesterase Inhibitors on Cognitive Function in Mild-to-Moderate Dementia: A Study Using Real-World Data"

_jcm, 2022, doi:10.3390/jcm11164661_

Round 1
Reviewer 1 Report
This manuscript observed the effectiveness of cholinesterase inhibitor 20 (ChEI) and nootropics combination in the treatment of cognitive impairment in dementia. It had a directive function for the treatment of dementia. The manuscript need minor revised.
1. The effects of choline alfoscerate, ginkgo biloba extrac tand other nootropics were discussed. But the authors don’t given the details of other nootropics. The details of other nootropics should be given in the maunscirpt.
2. The n value should be given for the ChEI only group and the ChEI + nootropics group in table 3.
Reviewer 2 Report
The authors report the analysis of real-world data of patients suffering from dementia by evaluating change in their cognitive functions in relationship to the medication they were prescribed. The object of analysis was the effectiveness of combination of cholinesterase inhibitor and one of several nootropics in the treatment of cognitive impairment in dementia. Health histories of 583 patients were analyzed providing quite a substantial data set for drawing conclusions. Though the number of patience in two analyzed groups (the one of patients treated just with cholinesterase inhibitor and the second one of patients who received a combination of cholinesterase inhibitor and nootropics) are quite different, but the distribution of patients in each of them in respect to other variables (mass, BMI, blood pressure, mean age, gender, dementia diagnosis, etc.) follow similar patterns what enables assumption that the observed tendencies in effectiveness of the treatment are reliable.
The results presented might be of interest for the search of the most effective combination for treatment of mild-to-medium dementia. The results of the analysis are presented clearly and logically. The discussion is comprehensive. The English language of the manuscript is suitable for the scientific text.
My minor comment is that the reference list does not completely meet the requirements of the journal – year must be bald and volume should be in italics.
